# A Broad-Spectrum Chemokine Inhibitor Blocks Inflammation-Induced Myometrial Myocyte–Macrophage Crosstalk and Myometrial Contraction

**DOI:** 10.3390/cells11010128

**Published:** 2021-12-31

**Authors:** Adam Boros-Rausch, Oksana Shynlova, Stephen James Lye

**Affiliations:** 1Lunenfeld Tanenbaum Research Institute, Mount Sinai Hospital, 25 Orde Street, Suite 6-1017, Toronto, ON M5G 1X5, Canada; adam.boros@mail.utoronto.ca (A.B.-R.); lye@lunenfeld.ca (S.J.L.); 2Department of Physiology, University of Toronto, Toronto, ON M5S 1A1, Canada; 3Department of Obstetrics & Gynecology, University of Toronto, Toronto, ON M5S 1A1, Canada

**Keywords:** uterus, labour, myometrium, broad-spectrum chemokine inhibitor, cytokines, chemokines, immune cells, infection, inflammation, TLR4, NF-κB

## Abstract

Prophylactic administration of the broad-spectrum chemokine inhibitor (BSCI) FX125L has been shown to suppress uterine contraction, prevent preterm birth (PTB) induced by Group B Streptococcus in nonhuman primates, and inhibit uterine cytokine/chemokine expression in a murine model of bacterial endotoxin (LPS)-induced PTB. This study aimed to determine the mechanism(s) of BSCI action on human myometrial smooth muscle cells. We hypothesized that BSCI prevents infection-induced contraction of uterine myocytes by inhibiting the secretion of pro-inflammatory cytokines, the expression of contraction-associated proteins and disruption of myocyte interaction with tissue macrophages. Myometrial biopsies and peripheral blood were collected from women at term (not in labour) undergoing an elective caesarean section. Myocytes were isolated and treated with LPS with/out BSCI; conditioned media was collected; cytokine secretion was analyzed by ELISA; and protein expression was detected by immunoblotting and immunocytochemistry. Functional gap junction formation was assessed by parachute assay. Collagen lattices were used to examine myocyte contraction with/out blood-derived macrophages and BSCI. We found that BSCI inhibited (1) LPS-induced activation of transcription factor NF-kB; (2) secretion of chemokines (MCP-1/CCL2 and IL-8/CXCL8); (3) Connexin43-mediated intercellular connectivity, thereby preventing myocyte–macrophage crosstalk; and (4) myocyte contraction. BSCI represents novel therapeutics for prevention of inflammation-induced PTB in women.

## 1. Introduction

Preterm birth (PTB) is defined as the delivery of a baby before 37 completed weeks of gestation and affects almost 10% of all pregnancies [1]. PTB is the major cause of death for neonates in high and low-income countries [1]. Depending on the study population, about half of spontaneous-onset (s)PTB is attributed to intrauterine infective processes with preterm premature rupture of membranes (pPROM) and half are of unknown etiology [2,3,4]. Intrauterine or systemic maternal infections that cause sPTBs can damage the fetus, leading to brain and lung injury [5,6,7]. Chorioamnionitis, local inflammation detected in the amniotic fluid, placenta, and fetal membranes/decidua, results in increased infiltration of maternal immune cells into the placenta and/or fetal membranes [8]. Uterine infection causing PTB could originate from ascending bacteria, particularly Gram-negative bacteria such as *Escherichia coli* (*E. coli*), as well as Gram-positive bacteria Group *B* *Streptococcus agalactiae* (GBS) [6,9,10].

The maternal immune system distinguishes infectious pathogen-associated molecular pattern (PAMP) molecules derived from exogenous foreign antigens, and sterile damage-associated molecular pattern molecules (DAMPs) derived from endogenous damaged/dying cells through specific ‘pattern recognition receptors’ called Toll-like receptors (TLRs) [11,12,13,14,15,16,17,18,19]. TLR engagement leads to the activation of intracellular pathways responsible for secretion of cytokines and chemokines, such as interleukin (IL)-8 (aka CXCL8), Monocyte Chemoattractant Protein-1 (MCP-1, aka CCL2), IL-1β, and IL-6 [20,21,22,23]. TLR transduction signals promote activation of Nuclear Factor kappa-light-chain-enhancer of activated B cells (NF-κB), a key transcription factor regulating expression of contraction-associated protein (CAP) genes by uterine smooth muscle (myometrium), such as Gap junction alpha-1 protein (*GJA1/*CX43); Prostaglandin F receptor (*PTGFR)*; Prostaglandin-endoperoxide synthase-2 (*PTGS2*, i.e., Cyclooxygenase-2); Oxytocin receptor (*OXTR)*; and pro-inflammatory cytokines such as *IL8*, *IL6*, *TNFα*, and *IL1β* [11,12,13,14,15,16,17,18,19].

Early reports from Norman et al. demonstrated the presence of physiological uterine inflammation during term labour (TL) [24]. Their data showed an increase in multiple uterine pro-inflammatory cytokines/chemokines (such as IL-1β, IL-6, and IL-8) during spontaneous TL [25], concomitant with infiltration of monocytes and neutrophils in human myometrium [24]. The myometrium secretes numerous cytokines and chemokines, which facilitate leukocyte migration in both TL and PTL [20,24,25,26,27,28,29,30,31]. Elevated local levels of intrauterine/intraamniotic pro-inflammatory cytokines and chemokines are thought to play a major role in PTB pathogenesis and fetal injury [21]. We recently reported that human myocytes isolated from myometrial tissue of term labouring women secrete higher levels of three cytokines (IL-8, MCP-1, and IL-6) in vitro as compared to myocytes isolated from term not in labour (TNL) myometrial samples [26]. The overwhelming evidence indicates that inflammation is detected in different uterine compartments (in particular, myometrium [20,24,25,26,27,28,29,30,31]) during PTL. Animal data showed that leukocyte infiltration into the pregnant rodent myometrium significantly increases during active labour with monocytes being a major contributor, exceeding neutrophils by three-fold [20,29]. Monocytes which leave the bloodstream and infiltrate into tissue can differentiate into macrophages, though ‘resident’ tissue macrophages already exist in almost every part of the body [32]. Macrophage function includes interactions with neighboring cells, creating the tissue microenvironment, migration to inflammation sites, antigen presentation, immunomodulation, as well as supporting physiological tissue functions [33]. The presence and infiltration of macrophages in the human uterus, as well as their contribution to local inflammatory signals culminating in labour onset has been well recognized and documented [34,35]. These immune cells attribute to 10% of the virgin murine uterine cell populations, and increase during pregnancy [34,35]. Furthermore, compared to women not in labour, women at term in labour or women in PTL display a high density of tissue resident macrophages [36]. Most significantly, it has been shown that macrophage depletion in a rodent model prevents infection-induced PTL [37,38] and cervical remodelling [38,39].

Our previous studies have demonstrated the feasibility of preventing PTB in mice [40] and non-human primates (NHP) [41] with a broad-spectrum chemokine inhibitor (BSCI). We have shown that the BSCI decreases in vivo infection-induced uterine inflammation by inhibiting chemokine-mediated monocyte and neutrophil infiltration into the myometrium [40], while in vitro, it prevents trans-endothelial migration of human leukocytes towards media conditioned by labouring myometrial cells [26]. More importantly, in pregnant monkeys the BSCI blocked pre-term myometrial contractions and decreased maternal plasma chemokine levels [41]. However, the molecular mechanism of BSCI action on uterine myocytes remains to be determined. Thus, we hypothesize that the BSCI directly inhibits myometrial inflammation by preventing chemokine secretion and CAP gene expression. We used human myocytes isolated from TNL pregnancies to study in vitro the effects of the BSCI on (1) activation of the NF-κB transcriptional pathway; (2) chemokine secretion; (3) CX43 protein expression and gap junction intercellular communication; (4) uterine myocyte contraction; (5) macrophage–myocyte communication. The results from these studies will provide important information regarding the molecular mechanisms of BSCI action in the uterus and inform the potential use of this or similar pharmacologic agents to prevent PTB in humans.

## 2. Methods

### 2.1. Ethics

All sample collections were approved by the Mount Sinai Hospital Research Ethics Board (MSH REB #04-0024E and #12-0007-E). Informed consent was obtained from all subjects involved in the study. Myometrial biopsies were collected from healthy term pregnant women undergoing elective caesarean section after receiving written consent. Peripheral blood was collected from third trimester pregnant women (37–40 weeks’ gestation) into EDTA vacutainer tubes.

### 2.2. Myometrial Cell Isolation and Culture

Myometrial biopsies were collected in 50 mL falcon tubes containing 25 mL of ice-cold HBSS with Ca^2+^ and Mg^2+^ (HBSS+/+) supplemented with 2.5% HEPES (Wisent, Montreal, QC, Canada) and 1% Penicillin/Streptomycin (Pen/Strep, Lonza, Basel, BS, Switzerland) Myometrial cell isolation protocol was based on techniques described by Srikhajon et al. [20]. Using small scissors and forceps, blood vessels were removed, and the myometrial tissues were cut into small pieces (approximately 1 mm^3^). Myometrial tissue was washed twice in HBSS+/+ to remove blood and twice in HBSS without Ca^2+^ and Mg^2+^ (HBSS−/−) supplemented with 2.5% HEPES (Wisent, Montreal, QC, Canada) and 1% Pen/Strep (Lonza, Basel, BS, Switzerland). Washing was performed by swirling tissue pieces in Erlenmeyer flasks containing 20 mL of buffer which was carefully aspirated between washes. After washing, enzymatic digestion solution containing: 10% fetal bovine serum (FBS) (Wisent, Montreal, QC, Canada), 1 mg/mL collagenase 2 (Sigma-Aldrich, Oakville, ON, Canada), 1 mg/mL bovine serum albumin (BSA) (Wisent, Montreal, QC, Canada), 0.15 mg/mL DNase 1 (Sigma-Aldrich, Oakville, ON, Canada) and 0.1 mg/mL Trypsin inhibitor (Sigma-Aldrich, Oakville, ON, Canada) was added to the tissue and placed in the rocking water bath (2 rpm) at 37 °C. Following 1 h incubation, the digested myometrial tissue was pipetted 30 times and passed through a 70 micron filter to collect a single cell suspension in ice-cold dissociation solution (HBSS−/−, with 10% FBS (Wisent, Montreal, QC, Canada), 1 mg/mL BSA (Wisent, Montreal, QC, Canada) to stop the enzymatic reaction. Undigested tissue was incubated with fresh enzymatic digestion solution to fully digest the remaining tissue (1 h in the shaking water bath at 37 °C) and the two cell suspensions were combined after the second round of digestion. The cells were centrifuged at 200× *g* for 10 min, washed with Dulbecco Modified Media (DMEM, Invitrogen, Carlsbad, CA, USA) containing 10% FBS (Wisent, Montreal, QC, Canada) + 1% Pen/Strep (Lonza, Basel, BS, Switzerland), and passed through a 23 G ¾ gauge needle. The cells were subsequently centrifuged at the same settings, resuspended in complete growth media (DMEM, 20% FBS + 1% Pen/Strep) and seeded in a 10 cm tissue culture plate (Eppendorf, Mississauga, ON, Canada). The cells were grown in a 20% oxygen incubator at 37 °C.

### 2.3. Myometrial Cell Treatment

Human myometrial smooth muscle cells were seeded into 6-well plates (passage 3–6, 100,000 cells/well) in DMEM (Invitrogen, Carlsbad, CA, USA) supplemented with 10% FBS (Wisent, Montreal, QC, Canada) and 1% pen/strep (Lonza, Basel, BS, Switzerland). When ~80% confluent, cells were serum starved for 24 h cell cycle synchronization using serum-free media (SFM, DMEM supplemented with 1% pen/strep and 1% ITS (Invitrogen, Carlsbad, CA, USA). Next cells were pre-treated with FX125L (250 nM [26]) or vehicle for 1 h followed by treatment with bacterial endotoxin, LPS (100 ng/mL, *E. coli*, O26:B6, Sigma-Aldrich, Oakville, ON, Canada) for 24 h.

### 2.4. Enzyme Linked Immunosorbent Assay

ELISAs were used to determine concentrations of specific cytokines in conditioned medium (CM) following various treatments. Human IL-6 and IL-8 ELISA kits were purchased from Thermo Fisher Scientific Inc (Waltham, MA, USA), human MCP-1 ELISA were purchased from R&D Systems Inc. (Minneapolis, MN, USA). CM were diluted 1:2 for IL-6 and IL-8 ELISA, using diluent solutions supplied by the manufacturer to ensure the absorbance reading would remain in range of the standard curve. For MCP-1 ELISA samples were used undiluted. Absorbance readings were conducted using μQuantTM software (BioTek^®^ Instruments, Winooski, VT, USA) according to the manufacturer’s instructions.

### 2.5. Monocyte Harvesting and Differentiation

Blood was collected from third trimester pregnant women (37–40 weeks’ gestation). Monocytes were isolated using the RosetteSep Human Total Monocyte Enrichment cocktail (STEMCELL Technologies, Vancouver, BC, Canada) following manufacturer’s instructions. Monocyte RosetteSep cocktail with EDTA was added to whole blood (50 μL/mL of blood), incubated for 20 min at room temperature, and diluted with an equal volume of PBS containing 2% FBS. This mixture was carefully layered on top of Ficoll (Sigma-Aldrich, Oakville, ON, Canada) and centrifuged at 1200 rcf for 20 min. Isolated monocytes were washed twice with 15 mL of PBS, resuspended in RPMI cell culture media supplemented with 10% FBS (Wisent, Montreal, QC, Canada), counted and plated at 5.0 × 10^6^ per T-25 flask (Corning, Scarborough, ON, Canada). Cells were allowed to differentiate for 10 days in media containing GM-CSF (100 ng/mL, Sigma-Aldrich, Oakville, ON, Canada), media was changed every 3 days and cells were visually monitored for morphological changes. Differentiated macrophages were harvested using enzyme-free Cell Dissociation Buffer (Thermo-Fisher Scientific Inc., Waltham, MA, USA), washed with RPMI supplemented with 10% FBS (Wisent, Montreal, CQ, Canada), pelleted at 200 g for 5 min, counted and plated as needed.

### 2.6. Immunocytochemistry

Isolated primary myometrial cells were seeded on glass coverslips, cultured for 48 h, and serum-starved for additional 48 h. For ICC co-culture, myocytes and macrophages were seeded at 1:100 ratio (macrophage to myocyte). These cells were then fixed with ice cold 50% acetone-50% methanol for 3 min, permeabilized with 0.02% Triton X (Sigma-Aldrich, Oakville, ON, Canada) and unspecific binding was blocked by 1% BSA (Wisent, Montreal, QC, Canada) in PBS−/−. For single staining myometrial cells were incubated with monoclonal rabbit anti-NF-κB p65 antibody (dilution 1:100, Cell Signaling, Danvers, MA, USA) overnight at 4 °C. After three washes, donkey anti-rabbit Alexa Fluor 594 was added and incubated for 1 h at room temperature. For dual staining to detect macrophage in co-culture, monoclonal mouse anti-CD68 antibody (dilution 1:100, Dako, Burlington, ON, Canada) in combination with polyclonal rabbit anti-CX43 antibody (dilution 1:100, Millipore, Etobicoke, ON, Canada) was incubated overnight at 4 °C. Goat anti-mouse (dilutions 1:300, Dako, Burlington, ON, Canada) and anti-rabbit Alexa 488 (1:1000, Sigma-Aldrich, Oakville, ON, Canada) were used as secondary antibodies. DAPI was utilised to stain the nuclei (dilution 1:1000, Sigma-Aldrich, Oakville, ON, Canada). Immunofluorescent images were captured using a Quorum Wave FX spinning disc confocal system comprising a Leica DMI 6000 B microscope (Leica, Wetzlar, HE, Germany) with a Yokoqawa Spinning Head and Image EM Hamematsu EMCCD camera and Velocity imaging software. To ensure fair comparison between donor and receiver cells in the parachute experiments, control images were taken first and time of exposure and laser intensity recorded and set. Treatment images were then captured using the same settings for each laser.

### 2.7. Protein Extraction and Immunoblotting

Cell lysates were prepared in lysis buffer (0.08 M Tris/HCl -pH 6.8, 2% SDS, 10% Glycerol) with freshly added protease and phosphatase inhibitor cocktail (Thermo Fisher Scientific Inc., Waltham, MA, USA) and total protein was extracted. Equal amount of protein was loaded to the gel with the first lane reserved for BLUelf Prestained Protein Ladder (GeneDireX, Taoyuan, Taiwan). Samples were separated by SDS-PAGE and transferred to a polyvinylidene difluoride (PVDF) membrane (Trans-blot Turbo Midi PVDF, BioRad, Mississauga, ON, Canada) using Turbo Trans-Blot system (Bio-Rad, Mississauga, ON, Canada). Membranes were incubated on a rocker for 1 h in blocking solution (5% milk in TBST or 1% BSA (Wisent, Montreal, QC, Canada) in TBST), followed by incubation with primary antibodies diluted in the blocking solution at 4 °C overnight (Phospho-IκBα (Ser32/36) and IκBα, Mouse monoclonal 1:500, (Cell Signalling, Danvers, MA, USA); Phospho-NF-κB p65 (Ser536) and NF-κB p65 Rabbit monoclonal 1:1000, (Cell Signaling, MA, USA); CX43, Rabbit polyclonal, 1:100, (Millipore, Etobicoke, ON, Canada). The membranes were washed thrice with TBS-T (for 10 min each) and subsequently probed with horseradish peroxidase-conjugated secondary antibody at room temperature for 1 h. Secondary antibodies for rabbit, mouse and goat were purchased from Amersham (1:5000). Before visualization via chemiluminescent detection, membranes were incubated with SuperSignal West Femto Maximum Sensitivity Substrate (Thermo Fisher Scientific Inc., Waltham, MA, USA) at room temperature for 5 min. All immunoblots were imaged on the ChemiDoc Imaging System (Bio-Rad, Mississauga, ON, Canada). To control for loading variations, membranes were stripped with Restore^TM^ PLUS Western blot stripping buffer (Thermo Fisher Scientific Inc., Waltham, MA, USA), and re-probed with anti-ERK2 antibodies (dilution 1:1000, Abcam, Waltham, MA, USA). Resulting images were scanned, analyzed and normalized to the housekeeping protein ERK2. Quantification of protein expression was conducted using Image Lab Software (Bio-Rad, Mississauga, ON, Canada).

### 2.8. ‘Parachute’ Dye-Coupling Assay

Gap junction intercellular communication (GJIC) was examined by the ‘Parachute’ dye-coupling assay, using two fluorescent dyes, CM-Dil and Calcein-AM as described in [42]. Myocytes were seeded in 6-well plates and grown to 50% confluency. Donor cells of each well were loaded with impermeable dye CM-Dil (5 μg/mL) and GJIC-permeable dye Calcein-AM (10 μg/mL) for 30 min at 37 °C, washed with PBS to remove unincorporated dyes, trypsinized and seeded onto a monolayer of receiver cells grown in another well. The ratio of donor to receiver cells was 1:100. To form GJICs between donor and receiver cells, co-cultures were incubated for 24–48 h at 37 °C. Immunofluorescent images were captured using a Quorum Wave FX spinning disc confocal system comprising a Leica DMI 6000 B microscope with a Yokoqawa Spinning Head and Image EM Hamematsu EMCCD camera and Velocity imaging software as described above.

### 2.9. Collagen Contractility Assay

Protocol was modified from Dallot et al. [43]. Briefly, confluent myocytes were trypsinized and re-suspended in DMEM supplemented with 10% FBS. A rat tail collagen type I solution (3 mg/mL in 0.1 N HCl, Thermo-Fisher Scientific Inc., Waltham, MA, USA) was adjusted to pH 7.2 with 0.1 N NaOH. Human myometrial cells at passages 3–6 were added to the neutralized collagen solution at a ratio of 1:1 for a total of 2 mL, the final concentration of collagen was 1.5 mg/mL. Collagen gel with cell suspensions was mixed in a 5 mL tube by inverting slowly until homogenized, and then carefully poured onto well (150,000 cells/well) of the 6-well culture plate (well diameter is 35 mm, Corning, Scarborough, ON, Canada), gently swirling to allow contact with plastic walls. For co-culture experiments, macrophages were also pretreated with FX125L (100 nM) 1 h prior to addition with myocytes in a ratio of 1:3 (macrophage to myocyte). Cell suspensions in collagen gel were incubated for 20 min at room temperature and then placed at 37 °C for 1 h to allow gelling. Subsequently, 2 mL of 10% FBS-DMEM was added over the collagen lattice. Two days later, the culture medium was replaced with SFM, and various test agents of interest were added to the SFM, including LPS (100 ng/mL). In some wells, FX125L (100 nM) was added 1 h prior to treatments with LPS. The lattices were then gently detached from the sides and lifted off the bottom of the culture dish. Images of the floating gels were captured at “time 0” before adding the test reagents and then every day for up to 5 days and digitized using a ChemiDoc Imaging System (Bio-Rad, Mississauga, ON, Canada); the area of the gels (mm^2^) were measured using Image J software. For each condition, collagen contraction was determined in quadruplicate (4 wells) at a minimum. Each experiment was performed at least 3 times using different primary myometrial cell lines (passage < 6). Results were expressed as mean gel area (mm^2^) ± the standard error of the mean (SEM).

### 2.10. Cell Viability Assay

Collagen gels were placed in centrifuge tubes containing 0.5 mL of a 0.1% collagenase solution (Sigma-Aldrich, Oakville, ON, Canada) and incubated for 10–20 min at 37 °C. Myometrial cells were pelleted by centrifugation at 200× *g* for 10 min and washed once with PBS−/−. Cells were stained with 0.4% Trypan Blue Solution (Thermo-Fisher Scientific Inc., Waltham, MA, USA) and viability was quantified using the Countess II Automated Cell Counter (Applied Biosystems™, Waltham, MA, USA). The measurement range extended from 10,000–10,000,000 cells/mL (with the optimal range: 1000–1,000,000 cells/mL).

### 2.11. Statistical Analysis

The normality of datasets was determined by the Shapiro–Wilk test. For the analysis of two groups, unpaired *t*-tests were performed for normally distributed data and the Mann–Whitney *U* test was utilized for nonparametric data, as specified in each analysis section. One-way ANOVA with Dunette’s post-test was employed to determine significance between data sets comprised of more than two groups (with normal distribution). The Grubbs’s outliers test was implemented with the assumption of a normally distributed population. Statistical analysis was performed using Prism 9 software (GraphPad Software Inc., San Diego, CA, USA). Significance level was set at *p* < 0.05 (*), *p* < 0.01 (**), and *p* < 0.001 (***).

## 3. Results

### 3.1. BSCI Suppresses LPS-Induced Chemokine Secretion by Human Myometrial Cells

Primary human uterine myocytes express multiple cytokines/chemokines in vitro [26], with MCP-1, IL-8, and IL-6 being the most highly secreted proteins. To determine the optimal time interval for induction of the immune response by bacterial stimuli, primary myometrial cell lines (*n* = 4–11) were stimulated with the TLR4 agonist LPS (100 ng/mL) or vehicle (SFM) for 8 or 24 h. Concentrations of MCP-1, IL-8, and IL-6 in CM were quantified by specific ELISA. Primary myometrial cells secreted significantly more cytokines in response to LPS treatment, as compared to vehicle. LPS-induced secretion of MCP-1 (370 ± 67 pg/mL vs. 685 ± 190 pg/mL, *p* < 0.001), IL-8 (332 ± 31 pg/mL vs. 760 ± 289 pg/mL, *p* < 0.01), and IL-6 (67 ± 32 pg/mL vs. 395 ± 141 pg/mL, *p* < 0.01) was significantly increased at 24 h compared to 8 h of stimulation (Appendix A). We concluded that stimulation for 24 h was sufficient to induce a strong immune response from human myometrial cells.

Next, human myocytes were pre-treated with BSCI (FX125L, 250 nM [26]) for 1 h followed by LPS (100 ng/mL) treatment for 24 h. Concentrations of MCP-1, IL-8, and IL-6 were quantified in CM by specific ELISA. BSCI caused a significant 3.7-fold decrease in inflammation-induced secretion of MCP-1 (318 ± 215 pg/mL vs. 87 ± 92 pg/mL, *p* < 0.05), and 2.7-fold decrease in IL-8 (565 ± 268 pg/mL vs. 213 ± 18 pg/mL, *p* < 0.05) (Figure 1). Notably, there was no effect of BSCI on LPS-induced secretion of IL-6 (Figure 1).

### 3.2. BSCI Significantly Inhibits LPS-Induced NF-κB Activation in Human Myometrial Cells

We next examined whether inhibition of LPS-induced myometrial cytokine secretion by BSCI was mediated through the NF-κB signaling pathway. Cultured primary myocytes were treated with LPS (100 ng/mL) or vehicle (SFM) with/out BSCI (FX125L, 250 nM) for 2 h and total protein was isolated. Western blot analysis was conducted to determine the phosphorylation of the NF-κB Inhibitor alpha (IκBα) and NF-кB p65 proteins. Treatment with LPS significantly induced phospho-IκBα (*p* < 0.001) and phospho-NF-кB p65 (*p* < 0.01) levels compared to vehicle indicative of NF-κB signalling pathway activation (Figure 2). Importantly, pre-treatment of myometrial cells with BSCI (FX125L, 250 nM) for 1 h significantly blocked inflammation-induced phosphorylation of both, IκBα (*p* < 0.01) and NF-кB p65 (*p* < 0.05) proteins (Figure 2B,C).

The activation of NF-κB signalling pathway by LPS was visualized via nuclear translocation of transcription factor NF-κB p65 (Figure 2D). Primary myocytes were treated with LPS (100 ng/mL) or vehicle (SFM) with/out BSCI for 2 h and immunostained with fluorescent-labelled anti-NF-κB p65 antibodies. Immunocytochemistry clearly demonstrated inflammation-induced nuclear translocation of NF-κB p65, as compared to its retention in the cytoplasm of untreated cells (Figure 2D). Furthermore, consistent with the immunoblot results (Figure 2A–C), LPS-induced NF-κB p65 nuclear translocation was blocked by BSCI (Figure 2D). Notably, pre-treatment of unstimulated myometrial cells with BSCI did not change their cellular localization (Figure 2D).

### 3.3. BSCI Inhibits Gap Junction Intercellular Communication between Myocytes

Next, we examined whether BSCI could prevent functional gap junction intercellular communication (GJIC) between myocytes. We first evaluated the presence of functional GJ channels between primary human myometrial cells using the ‘Parachute’ Calcein dye-coupling assay. As shown in Figure 3, under control conditions ‘donor’ myocyte cells (visualized by the red non-permeable dye DM-CiI directly loaded to the cells) were able to pass the fluorescent Calcein dye (Figure 3, green) to neighbor cells through open GJ channels. When ‘donor’ myocytes were pre-treated with the GJ inhibitor Carbenoxolone (CBX, 150 nM), GJIC was completely blocked (Figure 3). Similarly, pre-treatment of ‘donor’ myocytes loaded with DM-CiI with BSCI (FX125L, 250 nM) inhibited transfer of the Calcein dye between neighbor myocytes (Figure 3).

To further study if the blockage of the GJIC function by BSCI was due to its effect on the GJ protein CX43, primary myocytes were treated for 24 h with LPS (100 ng/mL) or vehicle with/out BSCI (250 nM), and CX43 protein was assessed by immunoblotting. No effect on the full length CX43 (CX43-FL) protein expression was detected; however, BSCI was able to downregulate the expression of the short isoform of CX43 protein (CX43-20 kDa) in both vehicle- and LPS-treated cells (*p* < 0.01 and *p* < 0.001, respectively, Figure 4A,B).

### 3.4. BSCI Inhibits LPS-Induced Myometrial Contraction

Collagen gel lattices were used to evaluate the effect of BSCI (FX125L) on inflammation-induced myometrial cell contraction. The contraction assay provided a functional assessment of the direct effect of the BSCI on myometrial contraction in vitro. Human myometrial cells, isolated from three different patients, were embedded in collagen gels, and stimulated with LPS with/out pre-treatment with BSCI. The contraction of collagen (measured as surface area of the gel) was recorded every 24 h. After 96 h of exposure of myometrial cells to LPS the percentage of gel contraction was 70%, while cells exposed to vehicle alone only exhibited a contraction of 26.2%, suggesting a direct action of LPS on myometrial cell contractility (LPS vs. control, *p* < 0.001, Figure 5A–C). When myometrial cells were pre-incubated with BSCI prior to LPS stimulation, we observed that LPS-induced myometrial contraction was fully inhibited (LPS vs. LPS + BSCI, from 70% to 24%, *p* < 0.01, Figure 5). Importantly, in the absence of LPS myometrial cells treated with BSCI showed minimal contraction, like that of untreated controls (SFM vs. SFM + BSCI, NS).

Given that BSCI blocked LPS-induced phosphorylation of transcription factor NF-κB activation in vitro in human myocytes, and inhibited myometrial gel contraction, we hypothesized that myometrial contractions are mediated by NF-κB activation. Therefore, we assessed whether inhibition of NF-κB activation could prevent LPS-induced contraction of collagen lattices containing myometrial cells. Primary human myocytes were pre-treated with/out the NF-κB inhibitor JSH (20 µM) for 1 h prior to stimulated by LPS. LPS-induced gel contraction was 35%, compared to vehicle-treated gel contraction of 9% (*p* < 0.01) after 24 h. Pre-treatment with JSH significantly inhibited inflammation-induced contraction of collagen gel (35% vs. 8%, *p* < 0.001, Figure 6). To exclude an effect of FX125L or JSH on cell viability, following termination of the experiments, gel lattices were treated with collagenase to recover and count cells. Results indicated similar cell viability between treatment groups, with no significant deviations (Appendix A).

### 3.5. BSCI Blocked Macrophage-Induced Myometrial Contraction

Since immune cells are involved in inflammation-induced myometrial activation in vivo [20] and interaction between macrophages and human primary cells is critical for LPS-induction of labour [37,38], we characterized the involvement of macrophage in myometrial cell contraction and the inhibition of this contraction by BSCI. Human macrophages, differentiated from peripheral blood monocytes of pregnant women, were co-cultured with myocytes isolated from term pregnant myometrium (ratio of 1:3). Macrophages were pre-incubated with/out BSCI (FX125L, 250 nM) before being added to collagen lattices embedded with human myocytes cultured in the continued presence of BSCI. The surface area of the gel was recorded every 24 h for 3 days. Macrophage–myocyte co-cultures induced significant contraction of collagen lattices as compared to myocytes cultured alone: after 72 h, the percentage of myocyte–macrophage co-culture gel contraction was 80%, while myocyte-only-containing gels contracted by 21% (*p* < 0.001, Figure 7). Inhibition of macrophage-induced myocyte gel contraction (co-culture vs. co-culture + BSCI, 80% vs. 17%, *p* < 0.001, Figure 7) was of a similar degree to that of BSCI inhibition of LPS-induced myocyte gel contraction (Figure 5). Notably, after 24 h human macrophages elicited significantly stronger contraction of collagen lattices containing co-culture of both cell types when compared to LPS-induced myocyte-only-containing collagen gels (27% vs. 62%, *p* < 0.05, Figure 5B and Figure 7B). 

We investigated potential mechanisms by which macrophage induced myocyte contractility. Notably, myocytes in direct contact with macrophage exhibited intense immunofluorescence when stained with anti-CX43 antibody, suggesting that the direct contact between these cells led to increased CX43 expression indicative of increased GJIC (Figure 8A). We then conducted a ‘parachute’ assay to confirm direct contact between macrophage and myocytes. When macrophage loaded with CM-DiI and Calcein-AM were co-cultured with primary human myocytes (macrophage-myocyte ‘parachute’ assay, ratio 1:100), there was a significant induction of GJ-permeable fluorescent green Calcein dye transfer after stimulation with LPS, as compared to unstimulated control cultures (Figure 8B). Figure 8B shows that LPS induces dye transfer from Calcein-loaded macrophages to adjacent myometrial cells, and that this transfer was fully blocked by the GJ inhibitor, CBX. Our results indicated that macrophages make GJ-dependent connections with neighboring myometrial cells. Finally, we investigated the effect of FX125L on the transfer of Calcein from macrophages to myocytes exposed to LPS, and found that, similar to CBX, FX125L prevented dye transfer from macrophages, possibly through inhibition of GJIC (Figure 8B).

## 4. Discussion

Two classes of chemokine inhibitors have been developed: (a) inhibitors specific for one or a small number of chemokine receptors, and (b) inhibitors that broadly impede the function of a wide range of chemokines (BSCIs) [44]. BSCIs are a very attractive therapeutic class of drugs in that in contrast to specific cytokine inhibitors, they simultaneously inhibit the actions of multiple cytokines, and possess wide-range anti-inflammatory function, similar to steroids but with reduced side-effects [44]. BSCI’s have been shown to mitigate the impact of inflammation in human and murine models of atherosclerosis [45], surgical adhesion formation [46], HIV replication [47], lung disease [48], endometriosis [49], and ischemia [50,51,52]. However, BSCIs do not antagonize the classical pathway involving chemokine binding to their receptors, but rather act as agonists by binding to cell-surface type-2 somatostatin receptors (SSTR2). Five sub-types of SSTR have been detected in human tissues [53,54]. SSTR2 and SSTR5 are highly expressed on inflammatory cells, including activated lymphocytes, monocytes, and endothelial cells [55,56]. The SSTR agonist, cyclic neuropeptide hormone somatostatin (SOM), mediates responses between the immune and nervous systems, regulating anti-inflammatory immune activity [57,58,59,60,61]. SSTR2 activation inhibits adenylyl cyclase and calcium entry by suppressing voltage-dependent calcium channels, which can affect several organ-specific functions [62,63,64,65]. Besides its role in regulating inflammation, recent studies show SOM and SSTR2 impact myometrial contractility [66]. SSTR2 expression was induced in porcine myometrial tissue infected by *E. coli* [66]. Administration of SOM increased the amplitude of porcine uterine contractions, while SSTR2 antagonists prevented contraction, suggesting a possible role for SSTR2-mediated uterine contractility [66]. We have recently shown that SSTR2 is expressed in the human myometrium [67].

In recent years, we performed several in vivo and in vitro studies to examine the potential of BSCIs to prevent PTL. Firstly, using pregnant mice as a model, we reported that the prototypic BSCI ‘BN83470′ reduced LPS-induced uterine inflammation, decreased concentration of pro-inflammatory cytokines in peripheral circulation, and in maternal tissues (myometrium, decidua, and liver), simultaneously inhibited leukocyte infiltration, and subsequently prevented inflammation-induced PTL [40]. Secondly, we showed that the BSCI compound FX125L (with superior pharmaceutical properties, including improved pharmacokinetics, safety, and toxicology), blocked myometrial contractions in a non-human primates (*Macaca nemestrina*) in which severe choriodecidual inflammation was induced by intrauterine instillation of Gram-positive bacteria (Group B *Streptococcus agalactiae*) [41]. Thirdly, we demonstrated the ability of a BSCI to block trans-endothelial migration of both human leukocytes (neutrophils, monocytes, and lymphocytes) isolated from the peripheral blood of pregnant women [26], and of the human monocytic cell line, THP-1 [28] when challenged by chemokines secreted by human myometrial cells (following stretch) or secreted by labouring human decidua and myometrium. Importantly, in both animal studies described above, the prophylactic administration of a BSCI was able to decrease myometrial chemokine release and contraction. In the current study, we sought to identify the molecular targets by which BSCIs inhibit myocyte contractility and prevent PTL.

Human myometrial cells are known to secrete abundant MCP-1, IL-8, and IL-6 proteins in response to the Gram-negative bacterial endotoxin, LPS [22,23,40]. Treatment of pregnant mice with LPS increases myometrial expression of *Ccl2* and *Cxcl1* and induces PTB, both of which are blocked by pre-treatment with BSCI [40]. In the current study, we found that FX125L was able to prevent LPS-induced secretion of two chemokines (MCP-1 and IL-8) from human myometrial cells, but not the secretion of pro-inflammatory cytokine IL-6. This is consistent with reports that activation of SSTR2 by specific ligands, prevents *CXCL8* and *CCL2* mRNA expression in LPS-activated human macrophages, suggesting its ability to interfere with TLR4 signaling [58]. Thus, we speculate that in myocytes BSCI binding to SSTR2 inhibits the intracellular signals initiated by TLR4, preventing LPS-induced chemokine secretion.

Murine labour is associated with TLR4 intracellular signalling involving the NF-κB transcription factor [68]. The activation of TLR4 on the surface of myometrial cells is essential for both TL and PTL, as TLR4 null mice exhibit delayed labour onset [69]. Importantly, human TL is also associated with an increased cytosolic and nuclear NF-κB p65 protein expression in the fundal region of myometrium [18]. Furthermore, activation of the NF-κB pathway in human myometrial biopsies is significantly higher in myometrium from TL compared to TNL samples [13,70]. A recent study by Chen et al. demonstrated that TLR4 expression in both labouring mouse and human myometrium was increased and positively correlated with expression of uterine CAPs (CX43, Oxytocin receptor/OTR, prostaglandin receptors), pro-inflammatory cytokines (TNF-α, IL-1β, IL-6), and chemokines (MCP-1) [13]. Furthermore, inhibition of the NF-κB pathway decreased the expression of inflammatory cytokines and chemokines as well as CAPs in human and mouse uterine smooth muscle cells [13]. Similarly, inhibition of TLR4 signalling by its antagonist naloxone, prevented intrauterine infection-induced (i.e., *E. coli*) PTB in mice [71]. In rhesus monkeys, TLR4 antagonism prevented LPS-induced preterm uterine contractility, as well as decreased prostaglandins and cytokine synthesis [72]. These data demonstrate a link between the induction of TLR4–NF-κB signalling and uterine contractility during labour [13,18,69,70,71,72,73]. In the current study, we showed that inflammation activates phosphorylation and nuclear translocation of NF-кB, while BSCI significantly blocked the NF-κB pathway in myometrial cells, likely preventing the expression of pro-inflammatory mediators. This provides a putative molecular pathway by which BSCI could impact LPS-induced chemokine secretion by human myometrial cells.

We previously reported that administration of BSCI (FX125L) completely suppressed myometrial contractility in a NHP model of preterm labour [41]. In the myometrium, individual cells are electrically coupled to allow elevated Ca^2+^ to spread between neighboring cells generating an action potential and muscle contraction. This intercellular communication occurs through molecular bridges formed by GJ proteins (connexins) allowing a passage of ions and certain small molecules. We speculated that BSCI may inhibit the expression or function of GJ proteins. CX43 is the most abundant connexin in the myometrium and its expression is increased in all mammals during TL and PTL [74]. While we failed to detect reductions in protein expression of the full length CX43 isoform (CX43-FL) following exposure to BSCI, the short CX43 isoform (CX43-20kDa) was significantly inhibited by this BSCI (the lack of effect of LPS on CX43 expression by these primary myometrial cells is likely due to their already elevated expression of CX43). The short CX43 isoform is responsible for chaperoning of CX43-FL to the plasma membrane to form functional GJ channels between neighbouring myometrial cells [75]. The myocyte–myocyte ‘Parachute’ dye-coupling assay confirmed that BSCI treatment decreased functional GJIC between cultured myocytes, possibly due to inhibition of CX43-20K-mediated transport of CX43-FL to the plasma membrane, which in turn would lead to reduced myocyte contractility and inhibition of PTL, as reported in our murine studies [40,41].

Collagen gels were previously used by Dallot et al. as a quantitative tool for measuring cellular contractility of myometrial cells [43]. We employed this strategy on cultured primary human myometrial cells and found that pre-treatment with BSCI prior to LPS insult significantly inhibited contraction of collagen gels, compared to LPS alone. These results demonstrated that FX125L was able to decrease inflammation-induced myometrial cell contraction which complements our published in vivo data [40,41]. To validate the hypothesis that TLR4-LPS stimulated myometrial contractions are blocked by BSCI through NF-κB-dependent mechanism, we utilized the specific NF-κB inhibitor, JSH [76] and found that inhibition of NF-κB translocation significantly decreased contraction of myocyte-containing collagen lattices challenged by LPS.

We and others reported that preceding labour, there is a significant increase in the number of uterine macrophages, though their exact role has remained largely unknown [20,24,27,34,35,36]. Multiple studies have shown that human monocytes/macrophages contribute to myometrial cell activation and contraction in vitro [77,78]. Rajagopal et al. demonstrated that monocytes infiltrating labouring myometrium participate in crosstalk that potentiate pro-inflammatory cytokine secretion [78]. More recently, Wendremaire et al. reported that macrophage-released ROS increases myocyte CX43 protein expression and contractility [77]. Our own studies have investigated the crucial role of monocytes during labour and their interaction with myometrial cells [20,29]. Since macrophages are involved in inflammation-induced myometrial activation in vivo [20] and myocyte–macrophage interactions are critical for LPS-induced labour mechanisms [37,38], we examined whether BSCI can influence macrophage-induced myocyte contractility and connectivity using in vitro co-culture of primary macrophages and myocytes. Prior research had mostly viewed macrophage-myocyte crosstalk in the uterus as being driven by paracrine stimulation. However, we observed that direct physical contact between human macrophages and myometrial cells (both derived from term pregnant patients) results in increased CX43 protein expression in myocytes. Hulsmans et al. recently discovered that cardiac macrophages electrically couple to cardiomyocytes through CX43-mediated GJIC to promote synchronized muscular contractions [79]. Using a ‘parachute’ assay, we were also able to show that macrophages communicate with uterine myocytes through GJs, and that this communication was induced by inflammation and blocked by BSCI. Further, we demonstrated that macrophage-myocyte co-culture caused contraction of collagen gels that was inhibited by BSCI. To our knowledge, our data are the first evidence that macrophage make GJ-mediated direct contact/coupling with uterine myocytes, and that this communication leads to increased expression of the CAP gene, CX43, and increased contraction of the myocytes. Given the key role of macrophages in labour onset, we propose that they may represent a target for PTB prevention by BSCI.

We recognize limitations of this research. Thus, bacterial endotoxin LPS used in vitro is an experimental tool to mimic infection-induced myometrial inflammation [40] and to examine the underlining mechanism of BSCI action; it does not necessarily recapitulate the full spectrum of inflammatory stimuli underlying the initiation of sPTB. Additional experiments using different PAMPs, or DAMPs are necessary to define the potential of BSCIs as therapeutics to prevent PTB.

In summary, our in vitro results demonstrate for the first time, the ability of the BSCI to block NF-κB pathway preventing inflammation-induced activation of human myometrial cells. We confirmed that LPS through TLR4 initiates intracellular signalling which activates (phosphorylates) NF-κB cascade and facilitates the transcription of pro-inflammatory proteins (IL-6, IL-8, and MCP-1) by myometrial cells. Our major findings are summarized in a putative model of BSCI’s action on myometrium (Figure 9). Thus, BSCI (1) inhibits TLR4-induced NF-κB intracellular signaling (likely acting through SSTR2) by blocking phosphorylation of the inhibitory protein IκBα, preventing phosphorylation of p65 protein and its translocation into the nucleus; (2) inhibition of this signalling pathway leads to decreased expression and secretion of myometrial chemokines; (3) through inhibition of CX43-20kDa expression, BSCI reduces CX43 trafficking to the membrane and thereby inhibits intercellular GJ communication, as well as (4) disrupting myocyte–macrophage interaction and inhibiting myometrial contractility. These actions contribute to the ability of this drug to prevent PTL in vivo in both rodent and non-human primate models of PTB.

## Figures and Tables

**Figure 1 cells-11-00128-f001:**
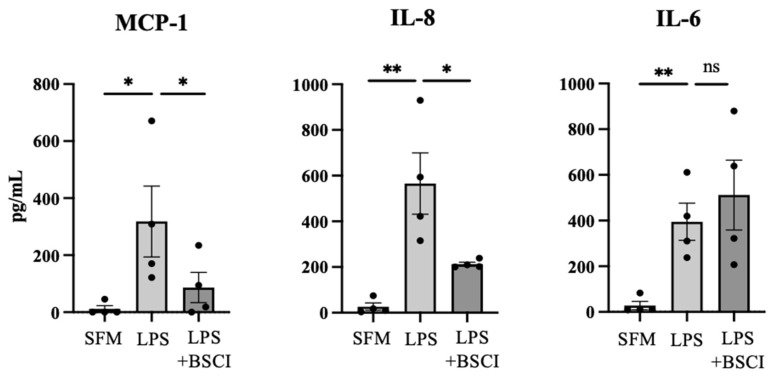
BSCI effect on infection-induced cytokine and chemokine secretion by human myometrial cells. Human cells isolated from myometrial biopsies of term pregnant women (*n* = 4) were pretreated with BSCI (FX125L,250 nM) for 1 h followed by treatment with LPS (100 ng/mL) or vehicle (serum-free media, SFM) for 24 h. Cell culture media conditioned by primary myometrial cells were collected and analyzed by specific ELISAs. Bar graphs show concentrations of secreted MCP-1, IL-8, and IL-6 in pg/mL. Data are presented as mean ± SD; dots represent individual patient cell lines. Statistical significance was determined by one-way ANOVA followed by Dunnette’s multiple comparisons test. ‘*’ denotes statistical significance at *p* < 0.05, and ‘**’ at *p* < 0.01.

**Figure 2 cells-11-00128-f002:**
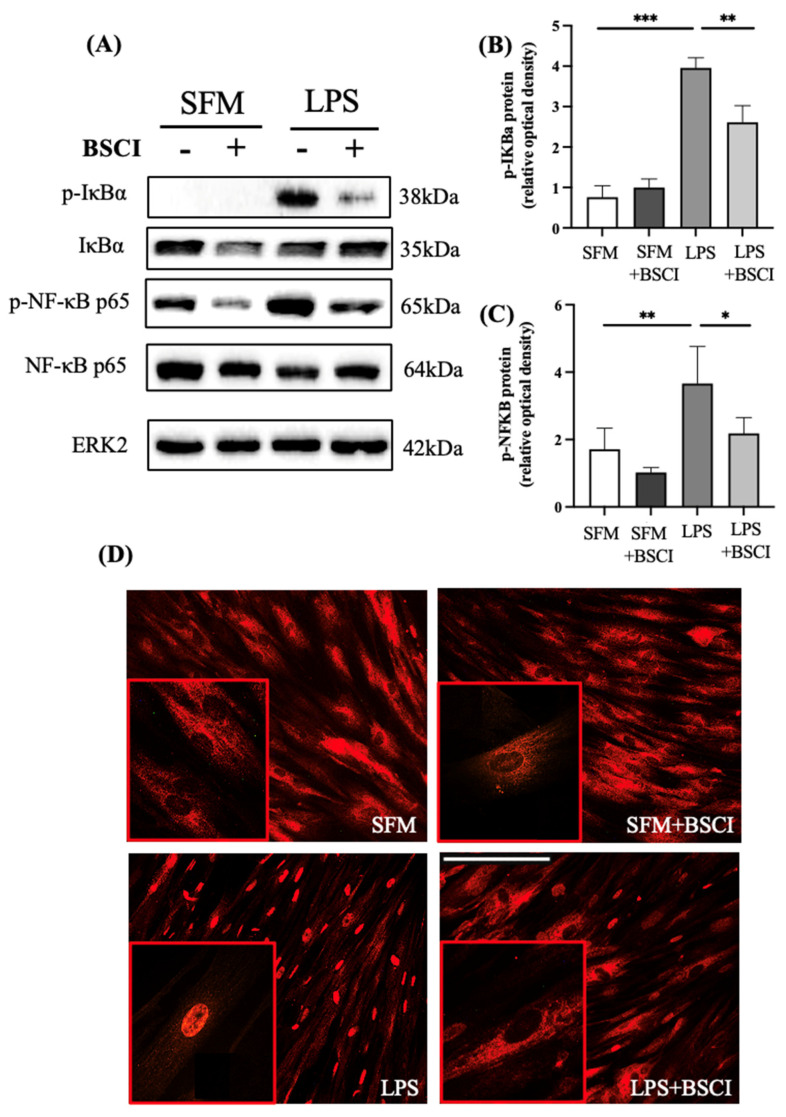
LPS activation of NF-κB intracellular pathway in human myometrial cells treated with BSCI. (**A**) Representative Western blot of human myometrial cells pre-treated with BSCI (FX125L, 250 nM), followed by treatment with LPS (100 ng/mL) or vehicle (SFM). Membranes were probed with anti-phospho-IκBα and anti-total-IκBα, as well as anti-phospho-NF-κB p65 and anti-NF-κB p65 antibodies, and normalized to ERK2 protein expression (loading control). Bar graphs show (**B**) phospho-IκBα and (**C**) phospho-NF-κB p65 for three individual cell lines (*n* = 3). Data are presented as mean ± SD. Statistical significance was determined through one-way ANOVA followed by Dunnette’s multiple comparisons test. ‘*’ denotes statistical significance at *p* < 0.05, ‘**’ at *p* < 0.01, and ‘***’ at *p* < 0.001. (**D**) Representative immunofluorescence images of human myometrial cells pre-treated with BSCI (250 nM) and treated with LPS (100 ng/mL) or vehicle (SFM). Cells were fixed and incubated with antibodies against NF-κB p65 (red stain, Alexa Fluor 594). Scale bar = 100 μm. Shown are images at 200× *g* magnification, inlet magnification 400× *g*.

**Figure 3 cells-11-00128-f003:**
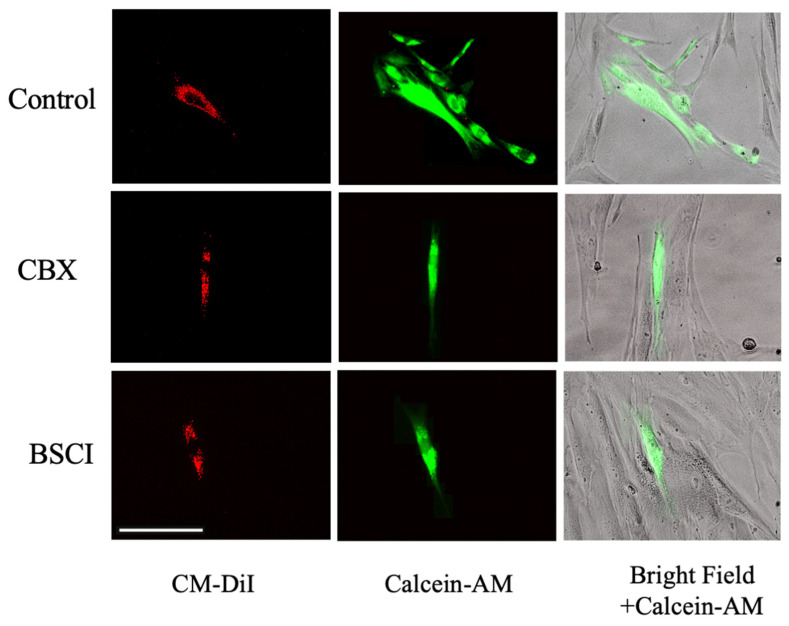
Myocyte-to-myocyte Calcein-AM dye transfer ‘parachute’ assay. Fluorescence images of myocyte ‘parachute’ assay: untreated myometrial cells (‘acceptor’) were grown in monolayer on glass cover slips; ‘donor’ myocytes were loaded with permeable Calcein-AM dye (10 µg/mL, green) and non-transferable cell tracker CM-DiI (5 µg/mL, red) and added in ratio 1:100. Cells grown in monolayer were incubated for 24 h with/out Carbenoxolone (CBX, 150 nM) or BSCI (FX125L, 250 nM), or Vehicle (Control). Representative images taken on a DMI spinning disc confocal microscope (Leica, Wetzlar, HE, Germany) identify ‘donor’ cells loaded with red CM-DiL; green fluorescent Calcein dye show transfer from ‘donor’ to ‘acceptor’ cells in Control; this transfer is fully blocked by CBX and BSCI. Scale bar = 100 μm. Magnification is 200× *g*.

**Figure 4 cells-11-00128-f004:**
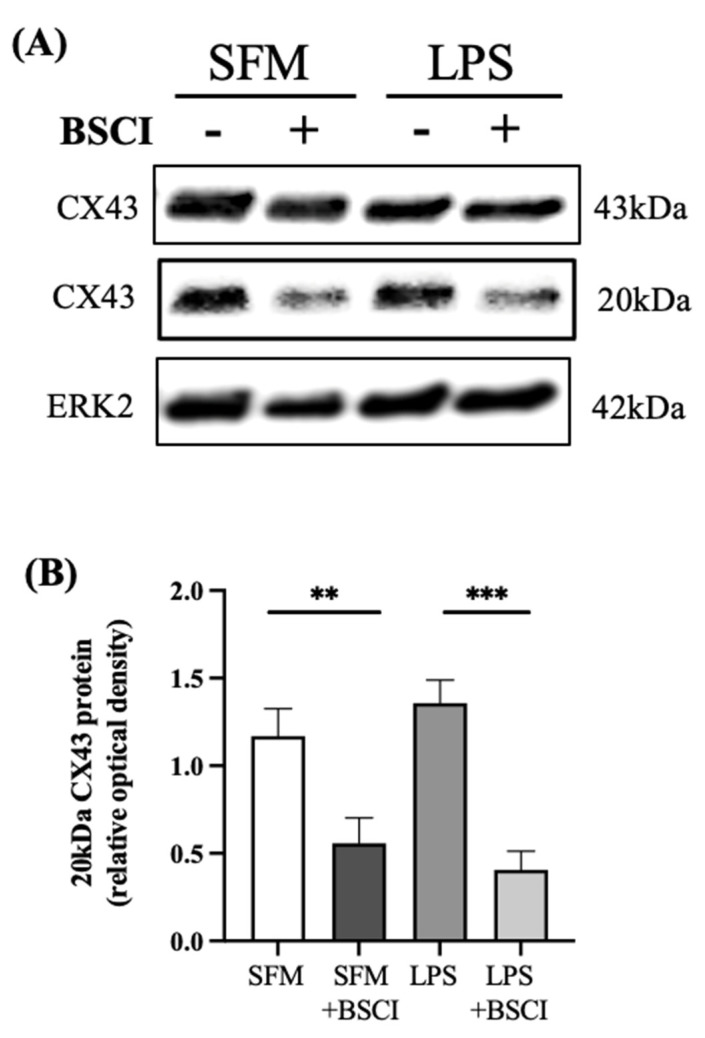
Effect of BSCI on protein expression of the gap junction protein CX43 in human myometrial cells. (**A**) Representative Western blot and (**B**) bar graphs showing the expression of the full length (43 kDa) and short isoform (20 kDa) CX43 protein, normalized to ERK2 protein (loading control) in human myometrial cell lines (*n* = 3) treated with LPS (100 ng/mL) or vehicle (SFM) with/out BSCI (FX125L, 250 nM). Data are presented as mean ± SD. Statistical significance was determined by one-way ANOVA followed by Dunnette’s multiple comparisons test. ‘**’ denotes statistical significance at *p* < 0.01, and ‘***’ at *p* < 0.001.

**Figure 5 cells-11-00128-f005:**
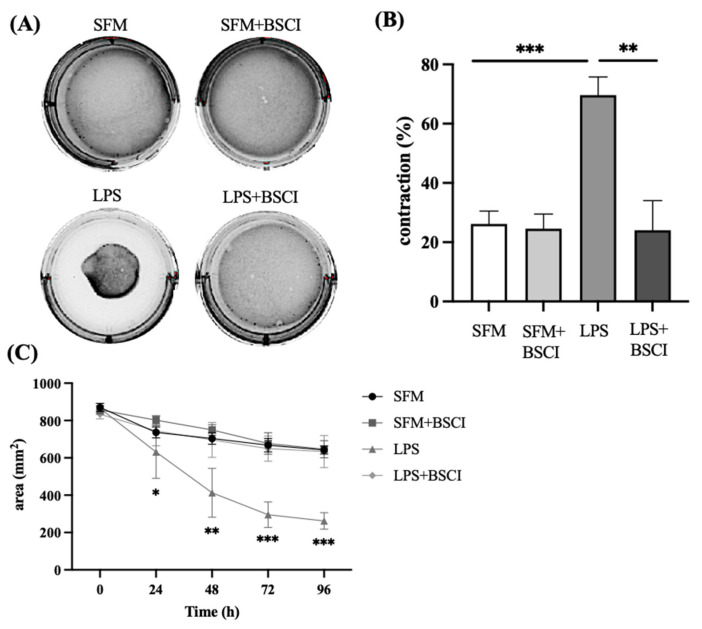
Effect of BSCI on LPS-induced myocyte–collagen gel contraction. (**A**) Photographs of collagen lattices containing human myometrial cells, induced with LPS (100 ng/mL) or vehicle (SFM) with/out BSCI (FX125L, 100 nM) for 96 h. (**B**) Bar graphs indicate the contraction of collagen lattices after 96 h of stimulation (% of the surface area of stimulated collagen gels at “96 h” compared to gels at time “0”). Data are pooled from *n* = 3 independent experiments. (**C**) Time-dependent changes in area of collagen lattices after LPS-stimulation with/out BSCI treatments. Individual human myometrial cell lines (*n* = 3) were used with *n* = 4 technical replicates for every experimental condition. Data are presented as mean ± SD. Statistical significance was determined by one-way ANOVA followed by Dunnette’s multiple comparisons test. ‘*’ denotes statistical significance at *p* < 0.05, ‘**’ at *p* < 0.01, and ‘***’ at *p* < 0.001.

**Figure 6 cells-11-00128-f006:**
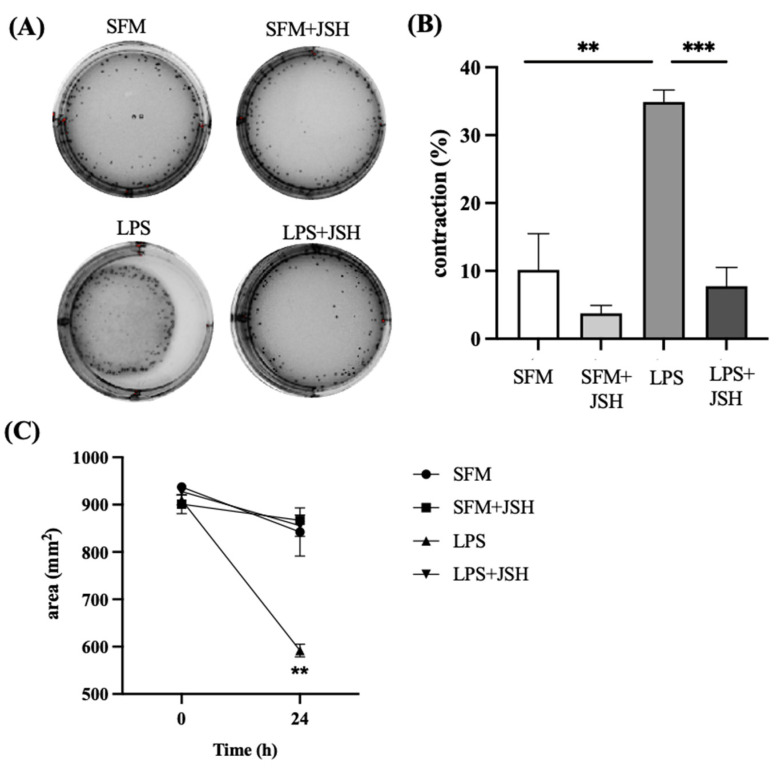
Effect of the NF-κB inhibitor JSH on LPS-induces myocyte–collagen gel contraction. (**A**) Photographs of collagen lattices containing human myometrial cells, induced with LPS (100 ng/mL) or vehicle (SFM) with/out JSH (20 µM). (**B**) Bar graphs represent the contraction of collagen lattices after 24 h of stimulation, compared to the basal surface area (percentage of gel area at time “0” versus “24 h”). (**C**) Shown are changes in collagen lattice area post LPS-stimulation with JSH treatments. Individual human myometrial cell lines (*n* = 3) were used with *n* = 4 technical replicates for every experimental condition. Data are presented as mean ± SD. Statistical significance was determined by one-way ANOVA followed by Dunnette’s multiple comparisons test. ‘**’ denotes statistical significance at *p* < 0.01 and ‘***’ at *p* < 0.001.

**Figure 7 cells-11-00128-f007:**
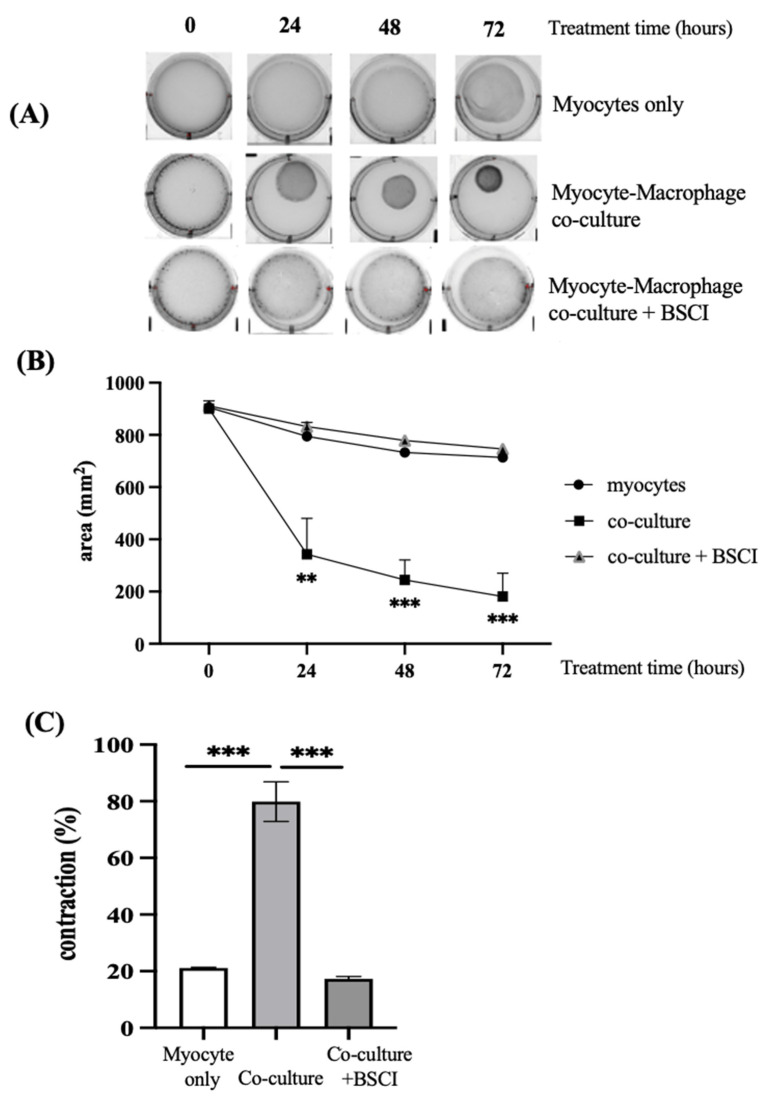
Effect of BSCI on macrophage–myocyte collagen gel contraction. (**A**) Photographs of collagen lattices containing primary human myometrial cells and blood-derived macrophages cultured for 72 h in collagen gels. (**B**) Time-dependent changes of collagen gel area after treatment with/out BSCI (FX125L, 100 nM). Human myometrial cell lines and macrophages were isolated from three term pregnant patients (*n* = 3), with *n* = 4 technical replicates per every experimental condition. (**C**) Bar graphs indicating the contraction of collagen lattices after 72 h (percentage of the collagen lattices surface area at time “0” versus “72 h”), in myocyte-only and myocyte–macrophage co-culture. Data are presented as mean ± SD. Statistical significance was determined through one-way ANOVA followed by Dunnette’s multiple comparisons test. ‘**’ denotes statistical significance at *p* < 0.01 and ‘***’ at *p* < 0.001.

**Figure 8 cells-11-00128-f008:**
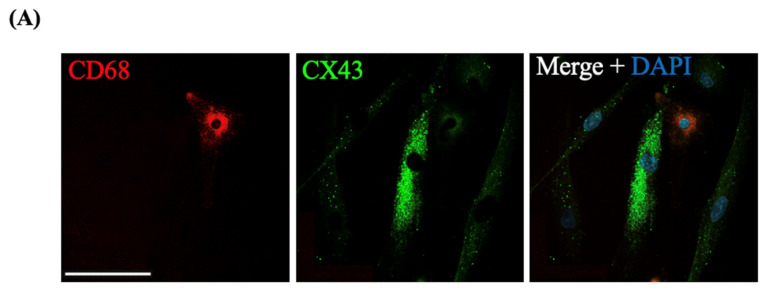
CX43 mediated GJIC between maternal macrophages and myocytes. (**A**) Representative immunofluorescence images of human myometrial cells co-cultured with blood-derived macrophages. Co-cultures were incubated overnight, fixed and immunostained with antibodies against CD68 (red stain, Alexa Fluor 594), CX43 (green stain, Alexa Fluor 488), and nuclear stain (blue, DAPI). (**B**) Macrophage–myocyte ‘parachute’ assay. Human myocytes (‘acceptor’ cells) were grown till 80% confluency and ‘donor’ macrophages were loaded with permeable Calcein-AM dye (green, 10 ug/mL) and non-transferable cell tracker CM-DiI (red, 5 µg/mL). The ratio of donor to acceptor cells was 1:100. Representative images show ‘donor’ cells loaded with red CM-DiL. Transfer of green Calcein dye from ‘donor’ to ‘acceptor’ cells after 24 h of incubation with/out LPS (100 ng/mL), carbenoxolone (CBX, 150 nM), and FX125L (250 nM) is apparent in Control and LPS-treated myocytes, the transfer is blocked by CBX and BSCI. All images were taken on a DMI spinning disc confocal microscope (Leica, Wetzlar, HE, Germany). Scale bar = 100 μm. Shown are images at 200× *g* magnification.

**Figure 9 cells-11-00128-f009:**
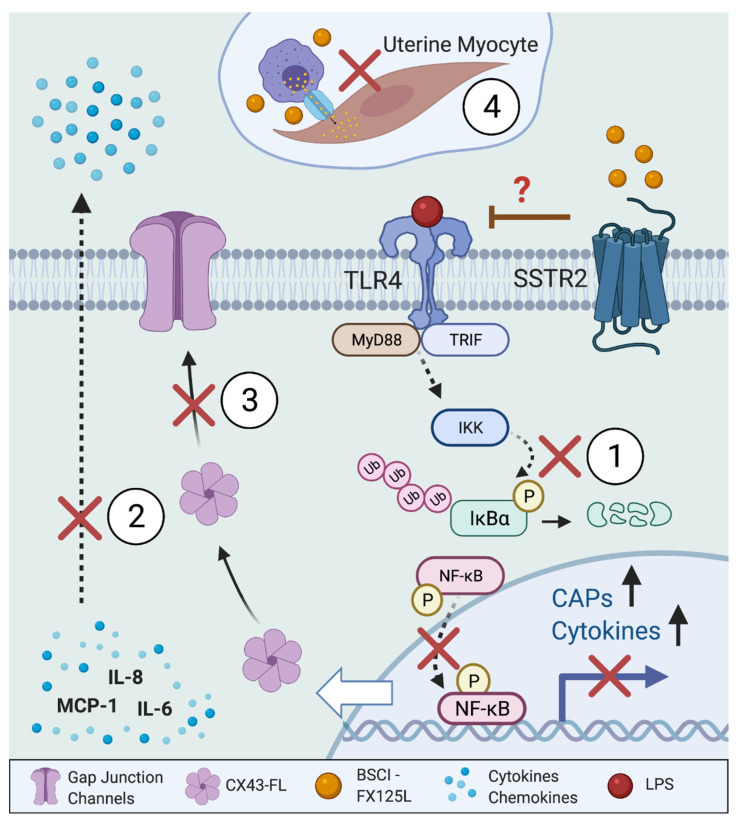
Putative model of BSCI’s action on myometrium. (1) BSCI (likely acting through SSTR2) inhibits TLR4-induced NF-κB intracellular signaling by blocking phosphorylation of the inhibitory protein IκBα, which prevents phosphorylation of p65 protein and its translocation into the nucleus, (2) thus decreasing the expression and secretion of myometrial chemokines, (3) CX43 trafficking to the membrane is decreased by BSCI, which inhibits intercellular GJ communication, (4) myocyte–macrophage interaction, and myometrial contractility.

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
