# Peer review of "A Broad-Spectrum Chemokine Inhibitor Blocks Inflammation-Induced Myometrial Myocyte–Macrophage Crosstalk and Myometrial Contraction"

_cells, 2021, doi:10.3390/cells11010128_

Round 1
Reviewer 1 Report
Boros & al studied the effect of BSCI, a chemokine inhibitor, on myometrial contractions, using a human experimental model. The experiments were performed correctly and the conclusion are in good agreement with the results. Nevertheless, there are some minor concerns that need to be answered.Minor comments :
- The difference between BSCI and FX125L is not clear. Please precise and use the same denomination throughout the manuscript.
Material and Methods
- the authors use a 1:100 ratio (macrophages to myocytes) for ICC (line 175) and a 1:3 ratio for collagen lattices (line 241) ? Why such as differences? What is the relevance to the pathophysiological context?
- In statistical analyses paragraph, the authors mentioned that they use a Bonferroni’s post-test (266) whereas in figure legends (lines 295, 323, 360…), they mentioned Dunnett’s post-test. Please clarify? In addition, if normality distribution test does not pass, which test did they use?
Results
- Line 387-388 : “…,we hypothesised that FX125L might inhibit NF-κB translocation.” This has already been shown in figure 2. Unless I'm mistaken, what the authors want to show on figure 6 is that myometrial contractions are mediated by NF-KB activation ?
Discussion
- The discussion part is a bit long, which has the consequence of weaken the results of the study, which are nevertheless very interesting.
- Line 508 : “Treatment” instead in “Treatmenbt”
- Line 617 : “FX125L” instead of “FX12L”
Author Response
We thank the Reviewer for his/her constructive comments on our manuscript. In particular, we appreciate the Reviewer’s recognition that our experiments were performed correctly, and the conclusion are in good agreement with the results.
The following issue have been corrected as recommended by Reviewer.
Question 1: The difference between BSCI and FX125L is not clear. Please precise and use the same denomination throughout the manuscript.
Response: We thank the Reviewer for this recommendation; we made multiple changes throughout the manuscript to accommodate this clarity.
Material and Methods
Question 2: the authors use a 1:100 ratio (macrophages to myocytes) for ICC (line 175) and a 1:3 ratio for collagen lattices (line 241) ? Why such as differences? What is the relevance to the pathophysiological context?
Response: We thank the Review for his/ hers comments. The 1:100 macrophage to myocyte ratio for ICC was based on the physiological presence of these resident immune cells in the human labouring myometrium as we and others have previously shown (Srikhajon et al, 2014, ref #20). Thus, we chose to use this ratio to appropriately examine their in vitro interaction by ICC approach. For the purpose of the ‘functional’ collagen assay, we used previously optimized method demonstrating contraction of collagen gel containing myocyte-macrophage co-culture (ratio 1:3), which was published in 2020 by Wendremaire et al. (ref #77). We recognize that this ratio in higher than physiological, detected in vivo in human tissue, but this was sufficient to demonstrate ‘proof of principle’ for this study, which was previously reported by others.
Question 3: In statistical analyses paragraph, the authors mentioned that they use a Bonferroni’s post-test (266) whereas in figure legends (lines 295, 323, 360…), they mentioned Dunnett’s post-test. Please clarify? In addition, if normality distribution test does not pass, which test did they use?
Response: We thank the reviewer for his/hers comment and have fixed the mis-typed error of Bonferroni’s post-test (line 266) which is now changed to the correct Dunnett’s post-test. To maintain normality, the Grubbs's test was implemented to detect outliers with the assumption of normally distributed population; however none of our data fell outside or failed normality distribution tests, thus this test was not reported. We will add this now to the text of statistics methodology for further clarification.
Results.
Question 4: Line 387-388 : “…,we hypothesised that FX125L might inhibit NF-κB translocation.” This has already been shown in figure 2. Unless I'm mistaken, what the authors want to show on figure 6 is that myometrial contractions are mediated by NF-KB activation ?
Response: We thank the Reviewer for this comment and corrected the text of the manuscript (Line 388)
Discussion
Question 5: The discussion part is a bit long, which has the consequence of weaken the results of the study, which are nevertheless very interesting.
Response: We thank the Reviewer for this comment and shortened the Discussion in few places.
Question 6: Line 508 : “Treatment” instead in “Treatmenbt”
Response: Thank you, we corrected the typo.
Question 7: Line 617 : “FX125L” instead of “FX12L”
Response: Thank you, we corrected the mistake.
Reviewer 2 Report
The study focuses on testing the mechanisms underlying the inhibitory effects of a Broad-Spectrum Chemokine Inhibitor on myometrial contraction related to infection-induced preterm birth. The authors provided consistent data supporting the authors' hypothesis in that the inhibitor FX125L prevent infection-induced contraction of uterine myocytes by inhibiting the secretion of pro-inflammatory cytokines, expression of contraction-associated proteins and disruption of myocyte interactions with macrophage via the NF-kB pathway. The data presented in this study confirmed previous studies showing a potential therapeutic benefits of the inhibitor in infection-induced preterm birth but offered new insights on the mechanisms regarding how the inhibitor exerts the beneficial effects, which is novel.
In general, the study is well-performed, data are properly analyzed and discussed.
Minor changes needed to clarify:
This is a clearly described study demonstrating a mechanisms of action and potential use of the Broad-Spectrum Chemokine Inhibitor (BSCI) FX125L in controlling LPS/infection induced myometrial contractility.
What is "Collagen43-mediated" in Abstract? Please check.
Author Response
In general, the study is well-performed, data are properly analyzed and discussed.
Minor changes needed to clarify:
This is a clearly described study demonstrating mechanisms of action and potential use of the Broad-Spectrum Chemokine Inhibitor (BSCI) FX125L in controlling LPS/infection induced myometrial contractility.
Question: What is "Collagen43-mediated" in Abstract? Please check.
Response: We thank the Reviewer for his/her constructive comments on our manuscript. In particular, we appreciate the Reviewer’s recognition that this work is important in offering new insights on the mechanisms regarding how the inhibitor exerts the beneficial effects in preventing Preterm Birth. The following issue have been corrected as recommended by Reviewer.
Changes are made in the text of the Abstract. We apologise for the oversight; it should be read: “Connexin43-mediated”.